# Realizing Hydrogen De/Absorption Under Low Temperature for MgH_2_ by Doping Mn-Based Catalysts

**DOI:** 10.3390/nano10091745

**Published:** 2020-09-03

**Authors:** Ze Sun, Liuting Zhang, Nianhua Yan, Jiaguang Zheng, Ting Bian, Zongming Yang, Shichuan Su

**Affiliations:** College of Energy and Power, Jiangsu University of Science and Technology, Zhenjiang 212003, China; 182210019@stu.just.edu.cn (Z.S.); 189210007@stu.just.edu.cn (N.Y.); 11526056@zju.edu.cn (J.Z.); tingbian89@just.edu.cn (T.B.); Zongmingy@just.edu.cn (Z.Y.)

**Keywords:** hydrogen storage, MgH_2_, Mn-based catalysts, catalytic effect, reversibility

## Abstract

Magnesium hydride (MgH_2_) has been considered as a potential material for storing hydrogen, but its practical application is still hindered by the kinetic and thermodynamic obstacles. Herein, Mn-based catalysts (MnCl_2_ and Mn) are adopted and doped into MgH_2_ to improve its hydrogen storage performance. The onset dehydrogenation temperatures of MnCl_2_ and submicron-Mn-doped MgH_2_ are reduced to 225 °C and 183 °C, while the un-doped MgH_2_ starts to release hydrogen at 315 °C. Further study reveals that 10 wt% of Mn is the better doping amount and the MgH_2_ + 10 wt% submicron-Mn composite can quickly release 6.6 wt% hydrogen in 8 min at 300 °C. For hydrogenation, the completely dehydrogenated composite starts to absorb hydrogen even at room temperature and almost 3.0 wt% H_2_ can be rehydrogenated in 30 min under 3 MPa hydrogen at 100 °C. Additionally, the activation energy of hydrogenation reaction for the modified MgH_2_ composite significantly decreases to 17.3 ± 0.4 kJ/mol, which is much lower than that of the primitive MgH_2_. Furthermore, the submicron-Mn-doped sample presents favorable cycling stability in 20 cycles, providing a good reference for designing and constructing efficient solid-state hydrogen storage systems for future application.

## 1. Introduction

Clean and sustainable energy is attracting tremendous attention worldwide because of the continuous shortage of fossil fuels and the worsening of environmental pollution. Hydrogen, which occupies higher energy density (142 MJ/kg) [1,2] than traditional fossil fuels and produces only clean and nontoxic water during combustion, is regarded as one of the most promising renewable energy resources [3,4]. Unfortunately, the utilization of hydrogen economy still faces many technical difficulties, especially for hydrogen storage [5,6,7]. Compared with the liquid and gaseous hydrogen, hydrogen stored in solid-state materials not only has the advantage of high hydrogen storage density, but also keeps safety during application [8,9,10]. Magnesium hydride (MgH_2_) with large mass hydrogen storage capacity (7.76 wt%), natural abundance, and excellent reversibility, ignites hope for meeting the demands of practical application of high-capacity hydrogen storage [11,12,13]. Nevertheless, its high thermodynamic stability and poor kinetic properties still lie in the way of practical application [14,15,16]. To conquer the above challenges, diverse technics like nanoconfinement [17,18,19,20,21,22], alloying [23,24,25,26,27], and catalyst doping [28,29,30,31,32,33,34,35] have been conducted over the past decades.

Transition metal halides were easy to be obtained and doped to MgH_2_ to improve its hydrogen storage properties [36,37,38,39,40,41]. Jangir et al. [42] observed that the initial desorption temperature of MgH_2_ was decreased by about 100 °C by doping TiF_4_ and the activation energy was lower by about 96 kJ/mol. Zhang et al. [43] successfully prepared a MgH_2_-NiCl_2_ composite and SEM tests exhibited that the addition of NiCl_2_ was conductive to decreasing the size of MgH_2_ grains and particles. Ismail et al. [44] doped FeCl_3_ into MgH_2_ to find that the desorption temperature of the MgH_2_-10 wt% FeCl_3_ composite was 90 °C lower than that of as-milled MgH_2_ and the activation energy for hydrogen desorption was also decreased from 166 kJ/mol to 130 kJ/mol. Mao et al. [45] revealed that the MgH_2_/NiCl_2_ sample could release 5.17 wt% H_2_ in 60 s at 300 °C and the dehydrogenation activation energy was decreased to 121.3 kJ/mol and 102.6 kJ/mol for MgH_2_/CoCl_2_ and MgH_2_/NiCl_2_ sample, respectively.

According to the above references, it can be concluded that doping transition metal halides into MgH_2_ could greatly enhance the de/hydrogenation properties. As far as we know, studies about Mn-based catalysts have rarely been researched, thus, it is urgent and interesting to explore the catalytic effect of MnCl_2_ for the reversible hydrogen storage performance of MgH_2_. However, there are still some shortcomings about the doping of transition metal halides. On one hand, the really doping amount of transition metal atoms are restricted because of the heavy halogen atoms. On the other hand, there would be a deadweight that Mg may react with halogen elements to form MgCl_2_ or MgF_2_ which could affect hydrogen capacity and absorption/ desorption rates [44].

In this work, the catalytic effect of MnCl_2_ was investigated and based on the microstructure evidence, submicron-Mn was successfully synthesized via a simple wet chemical method and doped directly to MgH_2_ to further enhance the hydrogen storage properties of MgH_2_. Moreover, its catalytic mechanism was explored and discussed in detail.

## 2. Materials and Methods

### 2.1. Sample Preparation

Powders of manganese chloride (MnCl_2_) was purchased from Sinopharm Chemical Reagent and Mn powders was commercially purchased from Aladdin Industrial Corporation. Submicron-Mn particles were prepared by a wet-chemical ball milling method. At first, 4 g Mn powders (99.95%, Aladdin Industrial Corporation, Shanghai, China), 12 mL heptane (98.5%, Sinopharm Chemical Reagent Co., Ltd., Shanghai, China), 0.6 mL oleic acid (90%, Sinopharm Chemical Reagent Co., Ltd., Shanghai, China), 0.2 mL oleylamine (98%, Sinopharm Chemical Reagent Co., Ltd., Shanghai, China), and 240 g balls were mingled in a home-made stainless steel jar under 0.1 MPa of Ar. The mixture was milled at a speed of 400 rpm for 60 h in the planetary ball mill (QM-3SP4, Nanjing, China). The treated slurry mixed with another 15 mL heptane, 1 mL oleic acid, and 1 mL oleic acid was then placed in a centrifuge tub. In addition, ethanol was used to centrifuge and wash the mixed solution eight times to remove larger particles and residual organic solvent. Finally, Mn submicron particles (submicron-Mn) can be acquired after vacuum-drying at room temperature for 12 h.

The MgH_2_ used was synthesized in our laboratory. First, Mg powder (99%, 100–200 mesh, Sinopharm Chemical Reagent Co., Ltd., Shanghai, China) was placed at 380 °C under the hydrogen pressure of 6.5 MPa to absorb hydrogen for 2 h. The second step was to ball-mill the processed samples at 450 rpm for 5 h. After repeating the above hydrogenation heat treatment, MgH_2_ can be finally acquired.

The MgH_2_ + *x* wt% submicron-Mn (*x* = 5, 10 and 15) and MgH_2_- MnCl_2_ composites were prepared by mechanical ball milling at 450 rpm for 2 h under 0.1 MPa of Ar (the ball to material ratio is 40:1). In order to avoid oxidation and contamination, all samples were handled and transferred in an Ar-filled glove box (Mikrouna, Shanghai, China) where the oxygen /water concentration was kept less than 0.1 ppm.

### 2.2. Sample Characterization

X-ray diffraction (XRD) analyses of all samples were performed on an X’Pert Pro X-ray diffractometer (PAN alytical, Royal Dutch Philips Electronics Ltd, Amsterdam, Netherlands) with Cu K alpha radiation at 40 KV, 40 mA to detect the phase compositions. To avoid air and water contamination, a special container was adopted for transferring and scanning samples. A scanning electron microscopy (SEM, Hitachi SU-70, Tokyo, Japan) with an energy dispersive spectroscopy (EDS) was performed to further characterize morphologies and element distribution of the samples. The hydrogen absorption and desorption properties were tested in a Sieverts-type apparatus. During testing non-isothermal hydrogen desorption properties, about 75 mg sample was heated to 450 °C at a heating rate of 2 °C min^−1^ in a sealed stainless steel reactor. For hydrogenation, the samples were gradually heated from room temperature to 400 °C at an average rate of 1 °C min^−1^ under 3 MPa H_2_. For isothermal measurements, the samples were first heated up to the desired temperature and then keeping the temperature constant in the whole test. In order to get the exact values of hydrogenation capacity, the second dehydrogenation measurements were also conducted to verify the accuracy of the values. Moreover, controlling the hydrogen pressure for de/hydrogenation tests well is also important, the isothermal absorption tests were performed at various temperatures under 3 MPa while the isothermal desorption performance was tested at different temperatures under hydrogen pressure below 0.001 MPa.

## 3. Results and Discussion

To investigate the catalytic effect of MnCl_2_ on the hydrogen storage properties of MgH_2_, 5 wt% MnCl_2_ was ball milled with MgH_2_ to prepare the MgH_2_ + 5wt% MnCl_2_ composite, and temperature-programmed desorption (TPD) tests were conducted from room temperature to 450 °C, the results are shown in Figure 1a. The un-doped MgH_2_ began to release hydrogen from 315 °C and about 7.45 wt% hydrogen could be desorbed after the non-isothermal test. It is clear that the dehydrogenation temperature shifts to lower temperature after doping MnCl_2_ powders, which started to release hydrogen at 230 °C and about 6.8 wt% hydrogen could be attained when heating up to 350 °C. XRD measurements were also used to shed light on the microstructure evolution in the desorption process. Figure 1b shows that MgH_2_ still dominates the diffraction peaks for MgH_2_-MnCl_2_ sample and no apparent new phases appeared during the ball-milling process, indicating that it was only a physical mixture after the ball milling process. After dehydrogenation, it is interesting that the signal of Mn phase emerged at 43° besides Mg phase. It is much likely that MnCl_2_ reacted with MgH_2_ during the dehydrogenation process, just as reported in other transition metal halides-modified MgH_2_ systems [42,44]. Therefore, Mn may be the key to enhance the dehydrogenation performance of MgH_2_. In order to confirm this conjecture, submicron-Mn particles were further synthesized and a series of tests were performed.

The as-prepared Mn particles were analyzed by XRD and SEM measurements, as shown in Figure 2. It can be seen from Figure 2a that the diffraction peaks in the XRD patterns of as-prepared Mn correspond well to that of the standard peaks of Mn phase (PDF#0637). Moreover, the diffraction peaks for as-prepared Mn become wider and weaker, suggesting a decrease in crystallite size under the effect of ball-milling. The SEM images from Figure 2b–d shows that the purchased Mn powders present the appearance of micrometer range particles while the particle size of as-prepared submicron-Mn ranged from 500 nm to 800 nm, which could be defined as submicron particles. Combing XRD results with SEM pictures, it can be concluded that submicron-Mn particles were successfully synthesized and a great catalytic effect was expected on the hydrogen storage properties of MgH_2_.

To witness the modification impact of the as-synthesized Mn submicron particles on MgH_2_, isothermal and non-isothermal dehydrogenation tests were operated. As a comparison, the purchased Mn was also doped into MgH_2_. The non-isothermal dehydrogenation curves in Figure 3a depicted that the onset temperatures of MgH_2_ + 5wt% Mn composite was 225 °C, 5 °C, and 90 °C lower than that of MgH_2_ + 5wt% MnCl_2_ composite and additive-free MgH_2_, respectively. Just as expected, after submicron-Mn was doped to MgH_2_, MgH_2_ + 5wt% submicron-Mn composite began to release hydrogen at 183 °C, superior to purchased Mn and MnCl_2_. In order to figure out the best doping amount, different amounts of submicron-Mn were ball-milled with MgH_2_ and further isothermal and non-isothermal measurements were conducted on the MgH_2_ + submicron-Mn composites. It could be clearly seen from Figure 3b that the volumetric release curves of submicron-Mn modified samples shifted toward lower temperatures with the increasing doping amount. The MgH_2_ + 5 wt% submicron-Mn composite possessed onset dehydrogenation temperatures of 183 °C, about 132 °C lower than that of prepared MgH_2_. As for the MgH_2_ + 10 wt% submicron-Mn and MgH_2_ + 15 wt% submicron-Mn composites, the onset desorption temperatures further decreased to 175 °C and 165 °C, respectively. When the temperature rose to 350 °C, about 6.8 wt%, 6.5 wt%, and 6.1 wt% H_2_ could be obtained for the MgH_2_ + 5 wt% submicron-Mn, MgH_2_ + 10 wt% submicron-Mn, and MgH_2_ + 15 wt% submicron-Mn samples, respectively. Further isothermal dehydrogenation measurements of the above three samples are performed at 275 °C. Figure 3c shows that only 4.7 wt% of H_2_ was desorbed in the first 10 min for the MgH_2_ + 5 wt% submicron-Mn composite. For the MgH_2_ + 10 wt% submicron-Mn and the MgH_2_ + 15 wt% submicron-Mn samples, the values increased to 6.1 wt% and 6.0 wt%. On the contrary, the pristine MgH_2_ sample could hardly release hydrogen under the same condition.

According to the results of TPD curves, it could be concluded that the addition of submicron-Mn could remarkably improve the hydrogen desorption kinetics of MgH_2_. Moreover, the initial dehydrogenation temperature did not decrease obviously after increasing the doping amount of catalyst. From a comprehensive perspective of the dehydrogenation temperature and capacity, the MgH_2_ + 10 wt% submicron-Mn was chosen for further study. Figure 3d presented the isothermal dehydrogenation curves of MgH_2_ + 10 wt% submicron-Mn composite at 250 °C, 275 °C, and 300 °C, respectively. The MgH_2_ + 10 wt% submicron-Mn composite could quickly release 6.6 wt% hydrogen in 8 min at 300 °C (almost 96.5% of theoretical hydrogen storage capacity). At 275 °C, this sample could desorb the same amount hydrogen with 20 min. Furthermore, about 6.0 wt% H_2_ could be acquired at 250 °C.

Besides the significantly improved dehydrogenation properties, hydrogen absorption kinetics of MgH_2_+ submicron-Mn composites were also investigated. As shown in Figure 4, the isothermal and non-isothermal hydrogenation tests were performed. From Figure 4a, it can be seen that the dehydrogenated MgH_2_ + 10 wt% submicron-Mn sample could absorb H_2_ even at room temperature and about 5.5 wt% hydrogen could be re-absorbed when heating up to 250 °C. However, the dehydrogenated MgH_2_ sample sluggishly took up hydrogen from 186 °C. The hydrogen absorption curves of MgH_2_ + 10 wt% submicron-Mn at relatively low temperature were also performed, shown in Figure 4b. Even at a low temperature of 50 °C, the dehydrogenated MgH_2_ + 10 wt% submicron-Mn sample still absorbed 1.8 wt% hydrogen within 40 min. When the temperature went up to 75 °C, the hydrogen uptake of the submicron-Mn-doped sample amounted to 2.3 wt% under the same condition. After hydrogenation at 100 °C for 40 min, the fully dehydrogenated composite could absorb 3.2 wt% hydrogen. As a comparison, non-isothermal hydrogenation measurement of MgH_2_ sample were also conducted (Figure 4c). For the dehydrogenated MgH_2_ sample, only 2.6 wt% H_2_ was absorbed even at 210 °C within 30 min.

In addition, the Ea values of the hydrogenation reaction were calculated to further explore the improved kinetics of hydrogenation for MgH_2_ + 10 wt% submicron-Mn sample. Some kinetic models such as Johnson-Mehl-Avrami-Kolmogorov (JMAK) model for the gas-solid reaction were adopted to simulate the evolution of kinetics [46,47]. Figure 4d depicts the JMAK model through fitting the absorption curves and the Ea values for the hydrogenation reactions were finally calculated according to Arrhenius equation [48].
ln[−ln(1 − *α*)] = *n*ln*k* + *n*ln*t*(1)
*k* = *A*exp(−*E_a_*/*RT*)(2)
where *α* is the fraction of Mg converted to MgH_2_ with time, *k* is an effective kinetic parameter, and *n* is the Avrami exponent. The numerical values of *n* and *n*ln*k* carried out by fitting the JMAK plots are shown in Appendix A.

In accordance with the curves shown in Figure 4d, the calculated Ea value of the hydrogenation process for the dehydrogenated MgH_2_ was calculated to be 72.5 ± 2.7 kJ/mol, while the value was reduced to 17.3 ± 0.4 kJ/mol for the dehydrogenated MgH_2_ + 10 wt% submicron-Mn sample. The greatly reduced activation energy also indicates that the energy barrier for hydrogenation was distinctly decreased after the addition of submicron-Mn, which well explains the evidently improved hydrogenation kinetics of the MgH_2_ + 10 wt% submicron-Mn sample.

Although doping submicron-Mn into MgH_2_ has shown great improvement in the reversible hydrogen storage properties, the catalytic mechanism of submicron-Mn in modifying MgH_2_ remained unknown. To further elucidate the hydrogen de/absorption mechanism, XRD tests of the MgH_2_ + 10 wt% submicron-Mn sample in ball-milled, dehydrogenated, and rehydrogenated state were performed. In the ball-milled state (Figure 5a), MgH_2_ phases still dominated the XRD pattern while the diffraction peaks of doped submicron-Mn were also found at around 43°. Interestingly, the XRD results demonstrated that the submicron-Mn was stable and persistently acted as an active substance during the process of de/hydrogenation. After dehydrogenation (Figure 5b) and 20th rehydrogenation (Figure 5c), the primary phase transformation during cycling is the transformation between Mg and MgH_2_. However, the diffraction peak of Mn which could be found at 43 ° was stable after 20 cycles and no other phases of Mn-related composites occurred, illuminating that Mn was the active catalyst to enhance the hydrogen storage properties.

In order to realize the practical application of hydrogen energy, preserving long-term kinetics is considered as one of the important indexes to evaluate the practicability for hydrogen storage materials. In this case, as shown in Figure 6, cycling tests of the MgH_2_ + 10 wt% submicron-Mn composite were further operated under the conditions of isothermal dehydrogenation and hydrogenation at 275 °C. As revealed in this pattern, the MgH_2_ + 10 wt% submicron-Mn sample could acquire a hydrogen capacity of 6.46 wt% in the first desorption. When exposed to hydrogen atmosphere of 3 MPa, the dehydrogenated sample quickly absorbed 5.94 wt% hydrogen. After 20 cycles, this composite could also release 5.72 wt% H_2_ (almost 89% of the original capacity). Compared with our previous study [29], the cycling property was better than that of MgH_2_ + nano-Fe samples, which had an evident decrease in the first 20 cycles. In general, the degenerating cycling properties caused by that MgH_2_ particles tend to grow and aggregate during the process of thermolysis [9,49]. Thus, although the addition of submicron-Mn can significantly enhance the de/hydrogenation performance of MgH_2_; other technics should still be explored to improve the cycling properties.

## 4. Conclusions

In summary, MnCl_2_ and Mn particles were doped as catalysts into MgH_2_ to improve its hydrogen storage properties and the submicron-Mn particles exhibited superior catalytic effect. The MgH_2_ + 10 wt% submicron-Mn composite started to release hydrogen at 175 °C and about 6.6 wt% hydrogen could be obtained within 8 min at 300 °C. For absorption performance, the dehydrogenated MgH_2_ + 10 wt% submicron-Mn sample began to absorb H_2_ at room temperature the completely dehydrogenated sample could assimilate 3.0 wt% H_2_ within 30 min under 100 °C while the dehydrogenated MgH_2_ needed a high temperature of 210 °C to absorb the same amount of H_2_. Besides, the E_a_ of hydrogen absorption of MgH_2_ was reduced to 17.3 ± 0.4 kJ/mol because of the addition of submicron-Mn. Moreover, the MgH_2_ + 10 wt% nano-Mn composite exhibited good cycling performance that 89% of initial hydrogen could still be maintained after 20 cycles.

## Figures and Tables

**Figure 1 nanomaterials-10-01745-f001:**
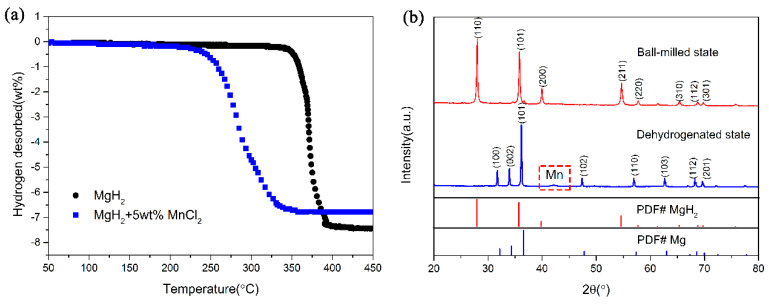
Volumetric release curves (**a**) MgH_2_, MgH_2_ + 5 wt% MnCl_2_ samples and XRD patterns of MgH_2_-MnCl_2_ composite in in ball-milling state and dehydrogenated state (**b**).

**Figure 2 nanomaterials-10-01745-f002:**
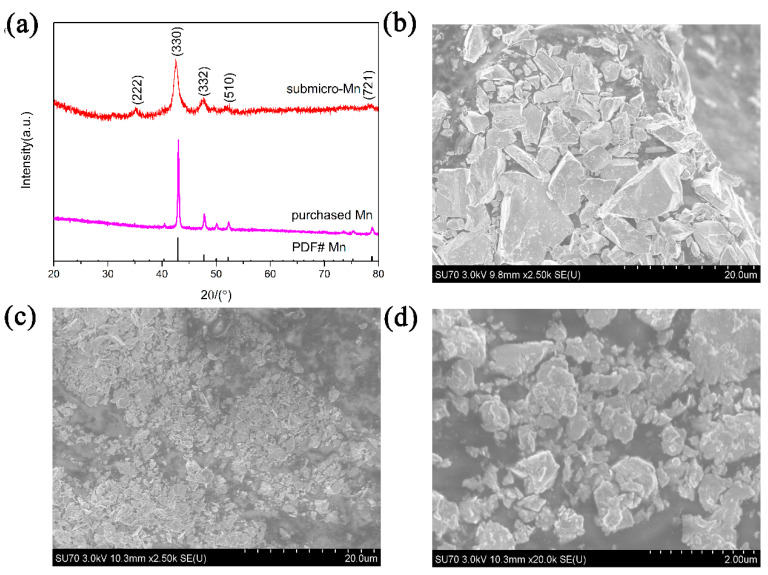
XRD patterns (**a**) and SEM images of purchased Mn (**b**) and as-prepared Mn particles (**c**,**d**).

**Figure 3 nanomaterials-10-01745-f003:**
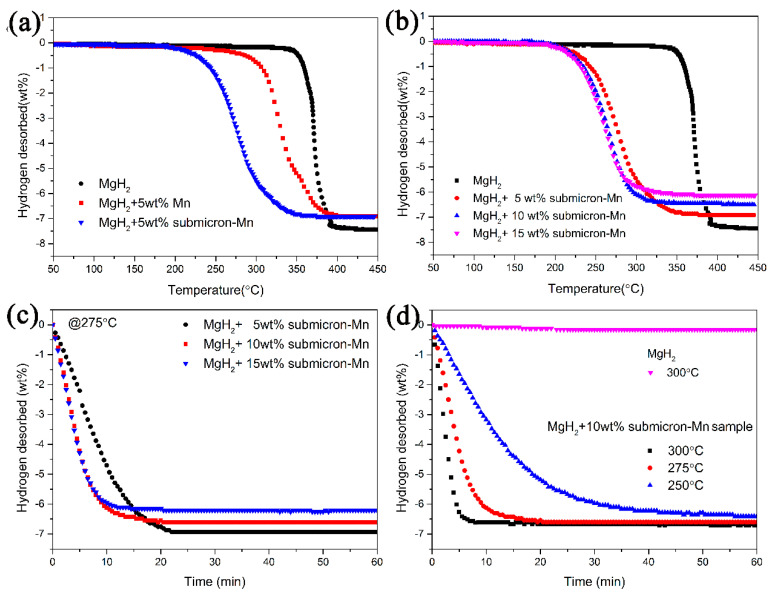
The volumetric release curves (**a**,**b**), isothermal dehydrogenation curves (**c**,**d**) of MgH_2_, MgH_2_ + Mn, and MgH_2_ + submicron-Mn samples.

**Figure 4 nanomaterials-10-01745-f004:**
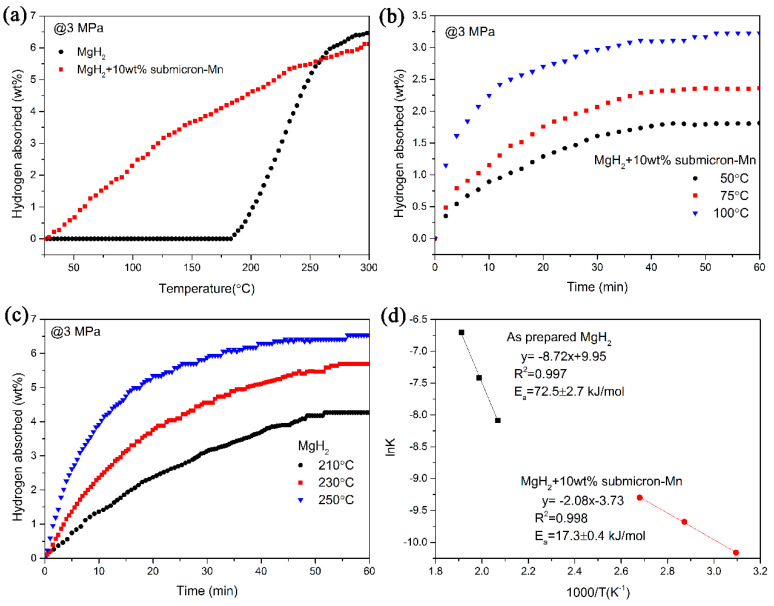
Non-isothermal hydrogenation curves (**a**), isothermal hydrogenation curves at different temperatures(**b**,**c**), and the relevant Arrhenius plots (**d**) of pure MgH_2_ and MgH_2_ + 10 wt% submicron-Mn sample.

**Figure 5 nanomaterials-10-01745-f005:**
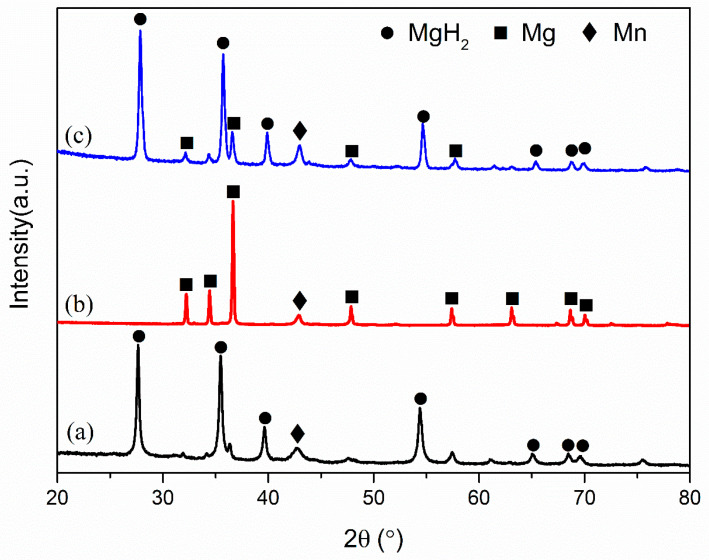
XRD patterns of MgH_2_ + 10 wt% submicron-Mn samples in three different states, ball-milled state (**a**), dehydrogenated state (**b**), and hydrogenated state (**c**).

**Figure 6 nanomaterials-10-01745-f006:**
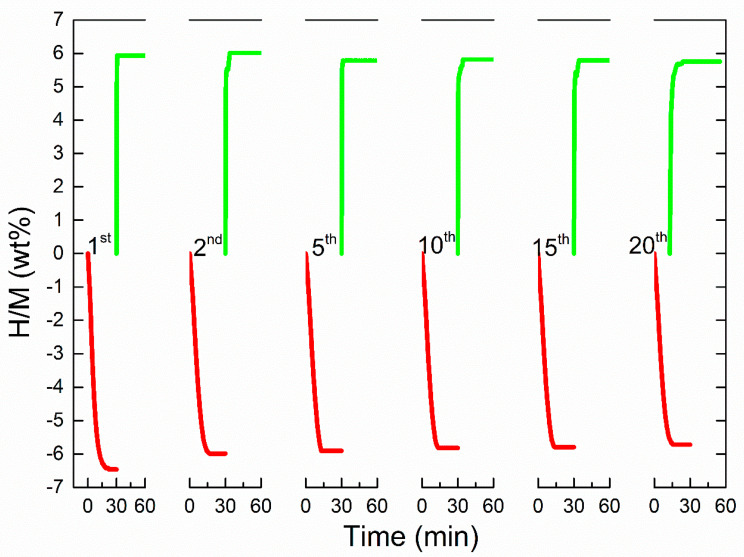
Isothermal dehydrogenation/hydrogenation cyclic kinetics curves of the MgH_2_ + 10 wt% submicron-Mn sample.

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
