# Peer review of "Realizing Hydrogen De/Absorption Under Low Temperature for MgH2 by Doping Mn-Based Catalysts"

_nanomaterials, 2020, doi:10.3390/nano10091745_

Round 1
Reviewer 1 Report
Dear Authors,
this manuscript describes “Realizing Hydrogen De/Absorption Under Low Temperature for MgH2 by Doping Mn Based Catalysts”. The subject of the work is of considerable importance and has been and is currently the subject of interest in the scientific fields, in particular in the hydrogen storage.
Small clarifications need for the work to be published.
In particular, the authors have to standardize the pressure units within the figures (MPa) and in the text (bars).
Furthermore, could the authors explain why they chose to study temperatures on the 10wt% Mn sample? There are no big gains in terms of H2 desorbed compared to 15wt% Mn but there could have been greater benefits at 300 °C. Are there possible explanations for this phenomenon?
Best regards
Regards
Author Response
Thanks for the reviewer’s comments. We have standardized the pressure units within the figures (MPa) and in the manuscript.
For increasing the doping amount of submicron-Mn, the dehydrogenation kinetics of MgH2 were gradually enhanced. Compared with MgH2+10 wt% submicron-Mn, the dehydrogenation temperature for MgH2+15 wt% submicron-Mn was not obviously decreased but the effective hydrogen content was decreased. Therefore, from a comprehensive perspective of the dehydrogenation temperature and capacity, the MgH2+10 wt% submicron-Mn was chosen for further study.
In this work, we performed the isothermal dehydrogenation tests for MgH2 +submicron-Mn samples at 275 °C. In our previous study[18], the isothermal dehydrogenation tests for MgH2 + Mn3O4 at 300 °C were conducted, what can be seen from Fig.X1 that the dehydrogenation kinetics were almost the same for the MgH2 + 5 wt% Mn3O4, MgH2 + 10 wt% Mn3O4 and MgH2 + 15 wt% Mn3O4 samples. In a word, the catalytic effect of doping catalysts could be better reflected under the condition of low temperature.
Fig.X1 Isothermal hydrogenation curves of MgH2 with and without Mn3O4 at 300 °C.
[18] Zhang, L.; Sun, Z.; Yao, Z.; Yang, L.; Yan, N.; Lu, X.; Xiao, B.; Zhu, X.; Chen, L., Excellent catalysis of Mn3O4 nanoparticles on the hydrogen storage properties of MgH2: an experimental and theoretical study. Nanoscale Advances 2020, 2, (4), 1666-1675.
We appreciate for reviewers’ warm work earnestly, and hope that the correction will meet with approval.
Once again, thank you very much for your comments and suggestions.

Reviewer 2 Report
The research paper describes the results of the study of the hydrogen sorption and adsorption temperature decrease for magnesium hydride by doping Mn based catalysts. The information in the paper is of high quality and importance for science. By combining the experimental studies and discussions, the obtained results were clearly described. The research paper is new, original and well organized. I recommend this paper for publication, after minor revisions:
- Figure 1 b should be improved by adding labels for all reflexes of XRD spectra.
- Figure 3 b is difficult to analyze due to wide lines.
- Further, all Figures are difficult to analyze in black and white format. Please make a different kind of points for different curves in the same Figure.
- MgH2 line should be added on Figures 3 b,c,d.
- What is the accuracy of determining the sorption characteristics? Please include this information in the research paper.
On the whole English language of the paper needs substantial improvement in grammar, scientific style in order to avoid the choice of high flown verbs such as deemed, e.g.
Author Response
- Figure 1 b should be improved by adding labels for all reflexes of XRD spectra.
Author reply:
Thanks for the kind suggestion. The labels for all reflexes of XRD spectra have been added in the manuscript.
- Figure 3 b is difficult to analyze due to wide lines.
Author reply:
Thanks for the reviewer’s suggestion. We have make some changes in this figure.
- Further, all Figures are difficult to analyze in black and white format. Please make a different kind of points for different curves in the same Figure.
Author reply:
Thanks for the reviewer’s advice. We have marked the different curves in several types to distinguish them.
- MgH2 line should be added on Figures 3 b,c,d.
Author reply:
Thanks for the reviewer’s good suggestion. We have added the MgH2 curve in Fig.3b and Fig.3d. For pure MgH2, we found that this sample could hardly release hydrogen even at 300 °C, so we did not put the dehydrogenation curve for MgH2 at 275 °C in Fig.3c.
- What is the accuracy of determining the sorption characteristics? Please include this information in the research paper.
Author reply:
Thanks for the comment. In our studies, we always conducted the second dehydrogenation measurements for these hydrogenation samples to verify the absorption amount. Moreover, some relevant discussion have been added in the manuscript.
For hydrogenation, the samples were gradually heated from room temperature to 400 °C at an average rate of 1 °C min-1 under 3 MPa H2. Isothermal measurements were operated by heating the sample to desired temperature and then maintaining the temperature constant throughout the whole test. In order to get the exact values of hydrogenation capacity, the second dehydrogenation measurements were also conducted to verify the accuracy of the values.
- On the whole English language of the paper needs substantial improvement in grammar, scientific style in order to avoid the choice of high flown verbs such as deemed, e.g.
Author reply:
Thanks for the kind suggestion. We have revised the whole manuscript and some high flown verbs were replaced.
We appreciate for reviewers’ warm work earnestly, and hope that the correction will meet with approval.
Once again, thank you very much for your comments and suggestions.
